# Speaking Stones: Oral Tradition as Provenance for the Memorial Stelae in Gujarat

**Durga Kale**

Department of Classics and Religious Studies (CLARE), University of Calgary, Calgary, AB T2L 2C3, Canada; durga.kale1@ucalgary.ca

**Abstract:** Anthropological fieldwork in rural settlements on the west coast of India has unraveled the close connection between lived experiences, spaces and objects. These "inalienable possessions", in the words of Annette Weiner, help reconstruct the past through the supplementation of oral traditions. Following this vein, the paper attempts to mesh together the material culture and oral histories to establish the provenance for the plethora of memorials in the state of Gujarat. A series of oral narratives collected in Western India since 2014 has highlighted the role of medieval memorial stelae that commemorate the deceased heroes of war and their wives and companions. This paper creates a niche for the Gujarati oral tradition as provenance for the continued veneration of these memorials. Field observations from 2014–2016 and notes from research in Gujarat from 1985 onwards enabled the study of patterns in the oral preservation of literature. A systematic documentation of the existing stelae and associated oral traditions has informed the views in this paper. The paper speaks to all levels of interaction and the making of an identity for the memorial stones that are unique to the state of Gujarat. A case for the inclusion of such rich material in museum displays is made in connection with this case study of the memorial stelae in Gujarat.

**Keywords:** oral tradition; vernacular sources of study; memorial stelae; Western India; material culture

## 1. Introduction

The semi-urban landscape in Gujarat along the Western coast of India is dotted with memorials that have been fashioned on local stone slabs since the medieval period in India (c. 800–1700 CE). These medieval memorial stelae, known as *palia*; commemorate the memory of deceased heroes of war and their wives or female companions who immolated themselves in the rite of "*Sati*". The rite mandated the immolation of the female dependents of the deceased warrior in some parts of medieval India [1]. The hero stones, locally known as *surapara*, and the *Sati* stones have come a long way since their initial execution as memorials in the medieval period.

Traditionally characterized by the horse rider and raised arm motifs (see Figure 1), these memorial stones occupy a status beyond mere "artefacts" in the present times. The memorials have also assumed the role of anchors for the repertoire of oral history in the region (Figure 2). Thinking of memorials as channeling the lived experience, Marius Kwint's succinct summary on the interaction of objects and humans becomes relatable [2]. Kwint positions the objects to serve memory in three main ways: They furnish recollection, stimulate remembering and form records. The second and third aspects in Kwint's thesis aid to enhance an understanding of human–object interaction at play in the case to be discussed. Objects such as these memorials form records: They are analogues to living memory, storing experiences beyond individual experience.

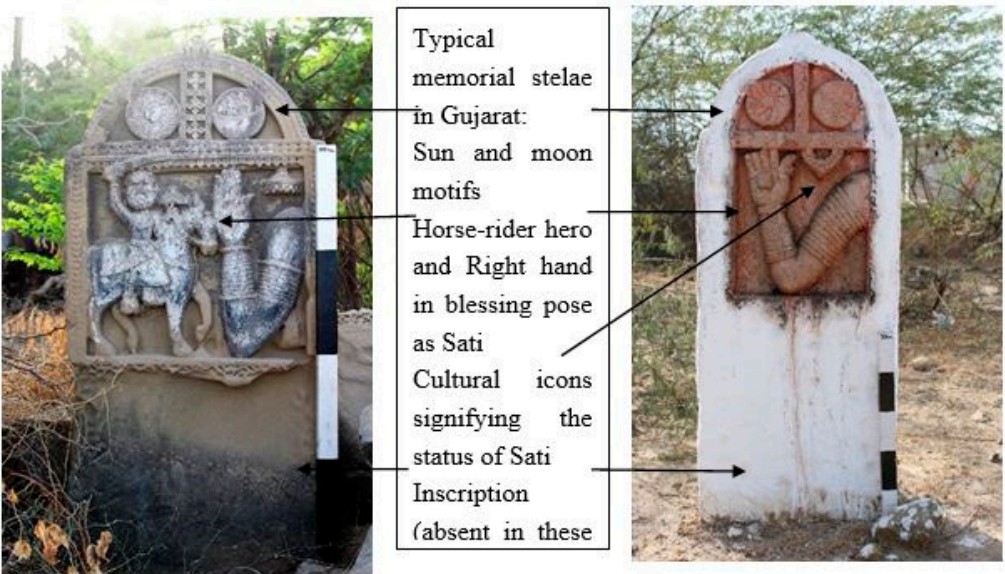

**Figure 1.** Memorial stelae in Gujarat with three sculptural components (all photographs by the author).

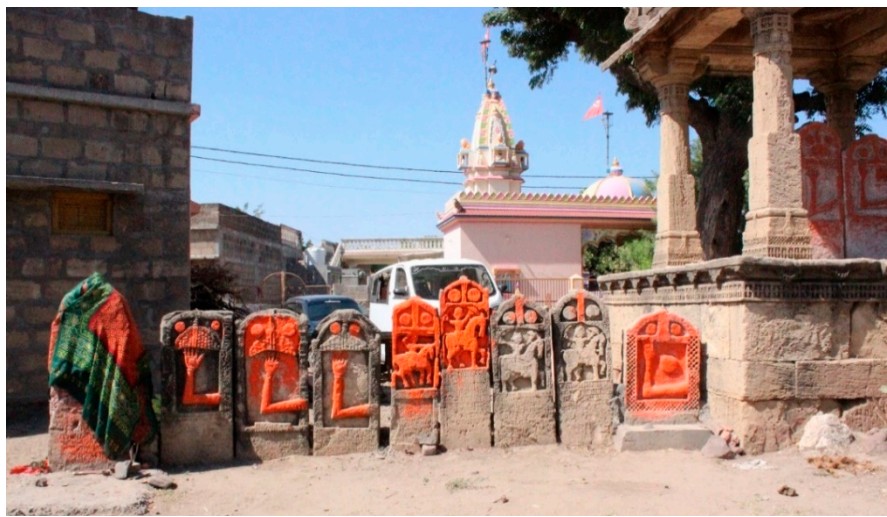

**Figure 2.** Memorials in the village of Chandroda. Note the vermilion used to promote the "deified" status, and the offering of saree on the Sati memorial (far left).

The paper develops Kwint's suggestions further, through the case example of the memorial stones in Kutch. Here, the "provenance" of the memorials derives its repertoire from the local narratives that describe the being of the stelae. Simply put, the immortalized hero and the local legends that elevate the ordinary looking slabs of stone to a culturally important piece of history serve as the source for the confines of this exercise that establishes the provenance. These records of memory are not synonymous with the provenance we are familiar with in terms of museum studies. The acquisition trail of an object in a museum forms the provenance. In the case of the memorial stelae, the acquisition, although important, real provenance comes from its surrounding cultural reception that makes these memorials cultural objects.

The following elaboration rides partly on the analysis of some theoretical concepts applied to the data collected during the fieldwork in the district of Kutch in Gujarat, and partly on the exploration of arriving at a provenance of thought and local performative traditions to elevate the hero and *sati* stones as cultural artefacts of Western India. The paper is divided into three main parts: The technical details of the fieldwork carried in Kutch, the richness of the local oral historical accounts surrounding the memorials and some suggestions for their presentation as part of the museum display of these memorial stones.

The hero and *Sati* stones housed in museums around the globe suffer from a lack of exhaustive presentation of the complex cultural backdrop that is offered by the original cultural setting of Western India (or any region based on the case, for that matter). A stelae standing in an exhibit, say at a museum such as the British Museum (Figure 3) manages to attract a nominal glance by patrons, who fail to associate it with the cultural milieu in which it was set. These stone stelae cannot compete with the elaborate and exquisite sandalwood carvings, nor can they tell their stories as beautiful artifacts made of jade or precious metals would, which have an inherent value. Perhaps museums diminish the experience of these memorials because a museum setting makes different aesthetic cultures compete, rather than reveal the "intrinsic" worth of memorial stones within its own cultural setting; which would happen were it is located in its native setting. With this perceived lacuna in the presentation of objects with insufficient source-literature until now, the purpose of documenting the Gujarat memorials was precisely to make the "stones speak", so that the history of polity and society would ensue from decoding these artifacts, which are the material anchors of effervescent lived traditions and history.

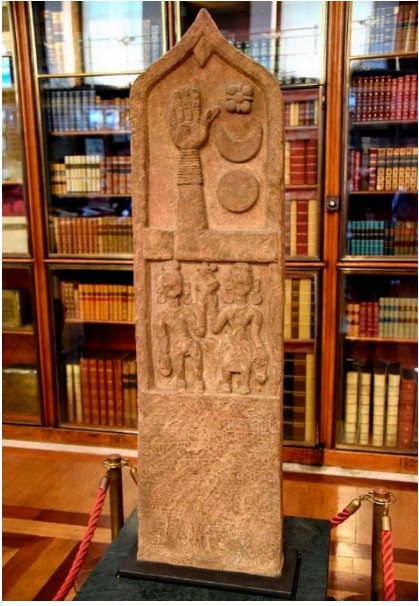

**Figure 3.** Sati stelae from Central India on display in the British Museum.

## 2. Goals and Methodology

The survey and fieldwork for this paper in the district of Kutch in Gujarat (India) uncovered some nuances pertaining to human–object interaction that Kwint revisits in his aforementioned thesis. A strong sense of legacy was also conveyed through the agency of worship and visibility of the lithic memorial stelae. The field trips to Kutch since 2014 have unraveled a plethora of living traditions and histories around the memorial stones. These traditions have become stepping-stones to further the study, analysis, and documentation of these memorials, some of which are housed in museums across the world. These oral histories and local traditions are indispensable given the lack of written sources concerning these memorial stelae. A general apathy for memorials of this nature on display in the museums is probably a reflection of the deficient research around these living objects of history.

This paper aims to reconstruct their provenance through material studies, such as archaeology and iconography, which are tied into the local oral histories that afford these material remains and give their due credit as historically important "artifacts" and anchors of living tradition in the area.

A major part of the fieldwork was conducted from 2015 to 2016 in the taluks of Anjar, Bhuj, Mandvi, Mundra and Nakhatrana of the Kutch district. The team of five researchers: Arya Peruvnath, Ranjini M, Renee, Heth Patel and the author started out by mapping important historical sites associated with wars and the tradition of memorials. The team extensively surveyed the village of Jara in Bhuj taluk[1] owing to its history as the site of the great war of Jara in the 18th century CE, which has immortalised numerous heroes in the form of memorials. The survey and documentation for this season of fieldwork aimed to develop a database of the extant memorials in the area. Geo-co-ordinates for the present location of the memorial were plotted, and the categorising of the memorial stelae as the hero or *Sati* memorial was completed. Documentation of oral narratives relating to the memorials was the primary focus of this project. Literature reviews conducted in the area from 1985 onwards (based on the last entry in the Gujarat State Gazette reports on religion and culture) consulted for the documentation have suggested several deified heroes and their companions rather than the narration of each martyr through the local narratives [3,4]. However, our village surveys revealed a sustained tradition of oral narratives for individual heroes. These narrativized accounts present the stories of each martyr along with the social significance in the village settlement pattern. This prompted us to further undertake a meticulous study of the narratives associated with the memorial stelae.

The data collection followed three stages: Background literary research and plotting the locations of memorials through already published datasets, field-visits, and later data analysis and storage in digital retrievable systems. On-field data collection involved the process of scaled photography, GIS plotting and ink-prints of the inscriptions as a means for studying the material sources. The ethnographic component of the data collection involved semi-structured interviews and recordings, the majority of which have already been transcribed. Following the guidelines from the CREB (Canada Research Ethics Board) and, abiding by the TCPS[2] (Tri-Council Policy Statement), the respondents were informed of the nature of the study and documentation. A systematic database with the details of the memorials and associated oral histories has been created and is maintained at the University of Kerala, Thiruvananthapuram campus. The database will be expanded to include data on the memorial stones from districts other than Kutch in Gujarat. The plan to document this tradition in the modern state of Gujarat is the overarching aim of the project.

The importance of oral narratives cannot be overstated in reconstructing the provenance of knowledge for living artifacts such as these. In the process of documentation, meticulous recording of oral narratives associated with the memorial stones in the study area were undertaken. This provenance will be indispensable in studying the present notions of religion and society in the area, as well as predicting some trends in socio-religious currents, even after the memorials have been moved from their find-spots.

---

[1] From; The Gazetteer of Bombay Presidency, Vol V: Kutch, Palanpur and Mahi Kantha. Bombay: Government Central Press, 1880.

[2] http://www.pre.ethics.gc.ca/eng/policy-politique/initiatives/tcps2-eptc2/Default/.

### 3. Material Evidence—Memorial Stelae

The memorial stones in Gujarat are generally executed as three horizontal bands. The main sculptural embellishment of the hero or the *Sati*, (or both, in the case of some examples) occupies the central register, while the upper band is embellished with the iconographic elements of sun, moon and floral motifs. A laudatory inscription is positioned on the third, lowermost register. The inscription, when present, informs the place of martyrdom and the date in *Vikram Samvat* (*Kartikadi* calendar: Year in *Vikram Samvat* − 57 = year in Gregorian calendar). Based on the numerous specimens studied in the district of Kutch, the lack of inscriptional evidence has added to complications in establishing the provenance. The same is true for most memorials of a similar nature in South Asia from the medieval period. That said, the provenance and origin stories of the memorials collectively and for individual stelae are fresh in the public memory. These memorials across the landscape in Gujarat are part of a social or public holding, almost functioning as "inalienable possessions" (in the words of Annette Weiner) for the locals [5]. The local population in surveyed villages view the memorials as a part of their heritage. Most of the memorial stelae are associated with some families, functioning as edifices for ancestor worship. Regular offerings and fresh coats of vermilion on the stelae are gestures of reverence for these memorials by the locals (as seen in Figure 2) [6].

Compared to the artifacts from the Indus Valley Civilization (c. 3300–1300 BCE) or other early civilizations in South Asia, the memorials in Kutch, Gujarat, represent an epoch closer to the lived present in the area [7]. Dated to just prior to the British colonial period in the subcontinent, these memorials render stories of brave ancestors—both male and female—whose deaths are seen as sacrifices for the motherland (here, respective villages and towns). We found the oral narratives for the memorials to be comprised of vivid details of war, the social standing of the hero and the devoted wife, and the manner in which the couple was deified upon their death. It is important to note that a majority of respondents were illiterate, and these narratives were passed on orally from the elders to the young in the family without the medium of written records. In an occupationally variegated society such as Kutch, the sociological impact of these memorials cannot be underplayed. These memorials often carry iconographic and/or inscriptional markers regarding the social group to which those portrayed belong. A memorial thus anchors the specific social group to that area in the village. After a family moves away, it is not unusual for them to undertake a kind of pilgrimage to visit the family memorial set in their original hometown in Kutch. Celebrated places of worship such as *Mata nu Madh* (lit. the cottage of the mother; which is now a representation of a Hindu deity but is believed to have originally been a Sati memorial), and temples in Kutch honoring specific family heroes have sprung out of this tradition of pilgrimage. This paper does not attempt to explore the traditions surrounding these places of worship, nor how the memorials have influenced the sociological makeup of the area, but it is important to note the ramifications of the popular worship received by the memorials, which are central to the rural landscape in Kutch.

Memorial stelae of the *Sati* and heroes can be found in museums across the globe. These artefacts on display, however, fail to communicate the cultural significance as stated previously. Often, visitors at a museum are not directed to look at artifacts of this nature. This largely results from the lack of accessible or targeted information on the cultural history of the specific object. In the case of these memorials, the sheer numbers in which they are found in their original setting attracts the eye and becomes the first imposing image. On the other hand, this very aspect might have been the reason for these stelae being understudied. The multiplicity of such memorials may have contributed to them being perceived as unimportant; a cautionary tale is provided by Davis, who points to the plethora of Hindu calendar art as being the reason for their having received miniscule attention for detailed research [8]. The second and most important factor for the reception of these objects is the trove of oral narratives that form the provenances of the stelae. Without these, the memorials would most likely be perceived merely as bland low-relief carvings on an ordinary-looking stone—slab. Greater stress on the exhibition-side of the stelae in museums is due, as it grants greater context to the artefact. Kerlogue mentions the "on stage" and "resting" lives of museum objects, where the objects on display

are the ones on stage [9]. Thus, the task for researchers is to make the objects speak, rather than just stand akimbo.

These memorials are a cultural cache and an indivisible part of the cultural landscape. It is important to point out here that in the planning of public works, etc. in the region, most of the schemes are planned 'around' the memorials, or a designated space within the village precinct is reserved for the memorials. The data from our fieldwork reflects the present locations of the memorials with respect to the village and temple (along with the geo-co-ordinates), which could be used for further study of cultural landscapes and so on. The data set is further studied to delineate nuanced tropes that emerged from this study. The memorials are a part of local living traditions, often styled as deities themselves and are worshipped for fertility and the protection of the village. Although every memorial is not necessarily unique, some memorials are embellished with specific iconography that draws its significance from the local belief system. It was therefore imperative to understand the connotations and symbology anchored in the community religion and belief system in the Kutch district.

## 4. Probing the Provenance

Various literary sources such as works by court poets composed around the medieval period in India (c. 800–1700 CE), following the tradition of *Pratimanatakam* of *Bhasa* (c. 300 CE) inform the reader about the practice of erecting memorials for deceased heroes who laid down their lives for the kingdom. Some later works, such as the *Chaturvarga Chintamani* by *Hemadri* mentions hero worship along with the rite of *Sati* for the female companions [10,11]. Several informed judgments about the nascent practice of this kind in Western India are extant in academic writing. A cursory look at the memorials in the region of Kutch would attest to the 'cookie-cutter' template followed for the hero and *Sati* stones, respectively. It is hardly surprising that the British colonial records halt at a nominal mention of these memorials given their non-particular appearance [12]. Our detailed study has revealed the nuances and individual narrative each memorial carries in the area. This was possible solely through local help with interpreting the cultural codes and peculiarities embedded in the rendering of these memorials. Examples of a nuanced understanding of the iconography of these stelae aided in visualizing the social context in which these artifacts appear, some of which are elaborated as follows:

- Jewelery represented in the depiction of the *Sati* memorial (Bangles on the raised arm of *Sati*): The local lores in Gujarat (and in the Kutch district) stresses the significance of multiple bangles worn by the bride. The ritual shattering or breaking of bangles upon the death of her husband signifies widowhood. The portrayal of the arms of *Sati* on the memorials accentuates her status as a prosperous woman entering the funeral pyre as a married person who would join her spouse in heaven. It is useful to point out that the status of being married is culturally held as the highest honour for a woman in Western India. As noted by Courtright in his work on iconography of *Sati* memorials; the aspect of the rite in which women accompanying their husbands in life and death is deified in this form [13].

- The cradle, umbrella/*chattra* and other symbols on the *Sati* stone: The accompanying symbology with the raised arm of *Sati*, such as a cradle, carries a cultural meaning according to the regional traditions. The cradle represents her status as a mother, in addition to being a devoted wife to the hero. The *chattra* or umbrella signifies her royal or respected status in the village. The flower motif or the symbology of the *haldi* and *Kumkum* casket is generally taken to signify the *Sati* as an elderly person at the time of her immolation.

- The hero on a camel: The selective representation of the hero on a camel instead of a horse was a puzzling aspect of the iconography. Some archaeologists and art historians believe it to be the choice of the one commissioning the memorial; however, a survey of numerous informants in Mundra revealed the selection of a specific animal as highlighting the status of the hero represented on the hero stone. The heroes depicted on camels were generally ministers or high-ranking officers of the army. Two cases of this have been attested to from the inscriptions on the memorials. In other cases, the inscriptions were either abraded or completely lost.

The exercise in documenting the provenance through oral narratives also presented a well-articulated backdrop of the kin relationships and, the power relationships between the rulers and foot soldiers, women in the *harem*, and those in the villages. Tapping into the regional oral histories has enabled the articulation of the unexpressed aspects associated with the memorials, bringing to mind Goffman's thesis on foreground and background actions, alluded to by Jhala in a similar study in Western India [14]. Goffman's thesis in sociology pertains to the actors' on-stage and off-stage performances. Jhala alludes to a similar notion of the memorial stelae operating as deified objects on one hand, and exuding power relationships with the village network, on the other. A dedicated thesis[3] to tease out these nuances is underway as a bi-product of our primary survey.

Furthermore, to establish the provenance from an archaeological perspective (the author's previous training), the issue of commissioning and executing the memorials was pertinent for the project. Local songs performed around the hero stones illustrate the process of championing the deceased person as a pious symbol, as well as how the community commemorated the memory. Susan Stewart (c.f Kwint 1999) noted that the poetics resemble a lyrical form, which is the verbal art of showing and sensing expression, thereby celebrating the encounter with the objects and their power to move, and summon something larger than perceived [15]. In this case, the larger picture is that of community reception and inter-relatedness through the medium of poetics surrounding the memorials. In some cases, through the oral narratives in Kutch, details such as the name and caste (occupational group to which the person belongs) of the mason, the name of the commissioner, the cost of commissioning the memorial, and so on, are woven into the poetic meter. Of more relevance to a project on history, these details of provenance are indispensable for the complete documentation of the material artifact under discussion. It can be argued that the oral transmission of this poetic lores ensures a stable form of retrieval system, with the agency of the bards.

The bards were traditionally trained poets and performers, generally holding their post as a hereditary village performer. The villagers continue to respect the bards and their families who have moved away from the traditional occupation. These bards and performers continue to carry information, though miniscule in comparison, regarding the historical details of *palia*. A suggestion that the ancient tradition of *phad* or painted screens which were used in public performances of narration by the bard were later transformed into durable pieces of stone sculptural panels. And the tradition of the memorial stelae emerged from the same. Traditionally, a narrative picture scroll formed the centrepiece of the *phad* performance, wherein sections of the picture are pointed out and the story of battle, conquest and valour is narrated through ballads as gathered from local respondents. Although shrouded in mystery regarding the origin and purpose for narrative performances, the memorial stones in Gujarat today continue to be the sites of *bhavai* performances by the bards. Tying into the tradition of *phad*, the singing of *bhavai* songs exists in the desert of Kutch in which lyrical compositions of valour of heroes and self-less sacrifice of *Sati*s are extolled through music, poetry and performance. The retinue would seat itself amidst the many memorials in respective villages (maybe a setting such as in Figure 4), and the bards would sing the *bhavai* songs re-enacting the battle scenes and act of sacrifice, often pointing at the carved stone slab to explain the nuances in its iconography.

3    Funded generously by the Department of Archaeology, University of Kerala (India) headed by Rajesh Sasidharan.

---

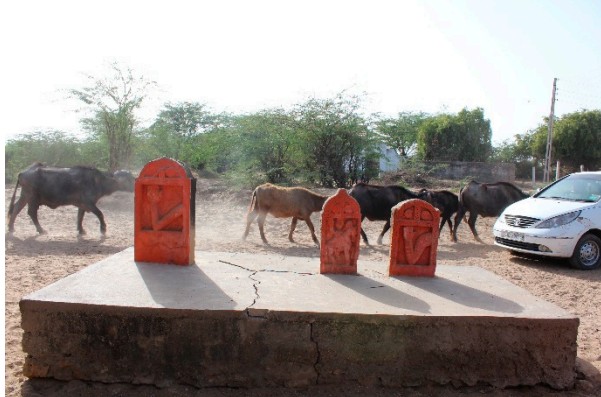

**Figure 4.** Platform for the performance of "*bhavai*" in the village of Bhada.

In light of a larger rubric, Kutch sits on the borderland between India and Pakistan, and has been the witness to many turbulent times in modern history, such as the Sindhi wars, Partition of India and the wars between India and Pakistan in the 1990s. The local lore of medieval memorials in the area incorporates a passing commentary on the brunt borne by the villages in these turbulent times. Recalling Jhala's (2017) study in Jhalawar (Central Gujarat), the empathic connections between the distant past and recent past are rejuvenated through the local narratives [16]. The present stratification in the society, hierarchy of occupational groups, and hereditary ownership of small stretches of land is thus mitigated through these narrativized histories that use the memorials as a conduit to link the past and the present.

The rehousing of memorials and continued engagement through commissioning new ones (as seen in Figure 5) highlights the importance of these memorials in Kutch. If memory only concerned past events, it would seem paradoxical that the local bodies of administration would pour their funds into such upkeep at the village level. Memory and memorialization in this case can actually be read as a concern for the present moment. Viaggiani illustrates a similar process in contemporary Northern Ireland of how collective memory and material culture are used to support the present political and ideological needs in contemporary society. Her investigation on how non-state organizations have filled a societal vacuum in the creation of public memorials is pertinent to the investigation at hand [17]. However, her further suggestion that groups sift through the past to propose "official" collective narratives of national identification, historical legitimation, and moral justifications for violence has yet to be met with in the case of the narratives in Gujarat. Jones, working on Neolithic material culture roughly aligns with the view of sustained dialogue between the past and the re-description of the present through material culture, while reiterating Merleau-Ponty's "reversibility thesis" [18]. The reversibility thesis lays out the position of a person in the world through the perception of the objects. It must be mentioned that such a "prescriptive perceptive" narrative may be in existence in Gujarat, as is currently discussed, but was not met with during the exercise of documentation. Without delving into these nuances, the mere mention of these aspects has been made, which will be examined independently in subsequent studies. With the changing political scenario globally, such cultural paraphernalia is often garbed as the desired narrative in the interest of groups in power. The case for a justifiable display of such artifacts in museums ensues from such palpable agency of material culture in the given social scenario.

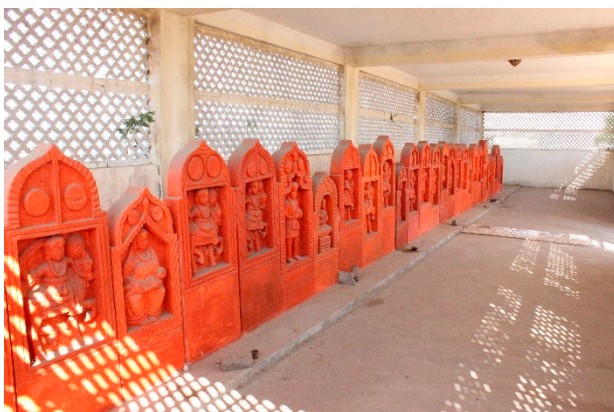

**Figure 5.** Congregation hall with modern replicas of memorials in the village of Bayath for public performance of *bhavai* songs and re-enactment of battle-scenes.

## 5. The Problem of Authenticity

In the documentation of cultural traditions such as these, the team faced the challenge of intelligibly assimilating the fluid lores and a plethora of metanarratives. The issue of "manufacturing of tradition" was also addressed to some degree. Selective social groups presented self-serving narratives with the deceased hero of the memorial as the beacon. As a study into the mentalities of the group, these "invented traditions", as coined by Eric Habswam, were incorporated into the documentation as an aspect of the continued tradition of re-description [19]. The insistence on the reception of these narratives was championed to highlight the continued interaction with the material culture in the form of memorials and local imagination.

In the process of establishing the provenance of epoch for the memorials through oral narratives, some hermeneutical methodologies were employed, to distinguish the trajectory of older to recent memorials and changes in the craftsmanship, art, and iconography. This was all undertaken to reiterate that these seemingly banal artifacts will be better understood and relatable with the correct provenance, which comes from its social setting.

The project has attempted to strike a balance between the raw data on field and its presentation in a culturally removed context. The map plotting and geo-co-ordinates for the memorials were assigned specific numbers in the database, and narratives were appended where relevant. The presentation of the excerpts from the oral accounts in a museum display for example, will either be through storyboards or as recordings, which will be accessible in the near future. The idea is to present a "sensorium" with the associated object. While standing in a village, these memorials are an integral part of lived experiences, with the associated speakers and informants at hand. The documentation undertaken for the said project is the first step in tapping into the auditory, visual, and remembered aspects pertaining to the memorials. Documenting the stories associated with the memorials was began with the project and continues today as a further step into audio and video recording of the *bhavai* songs performed in the vicinity of the memorials, discussed in Section 4. The museum displays could then induct these data sets into recreating a microcosm as the accompanying display with the memorial (Figure 6).

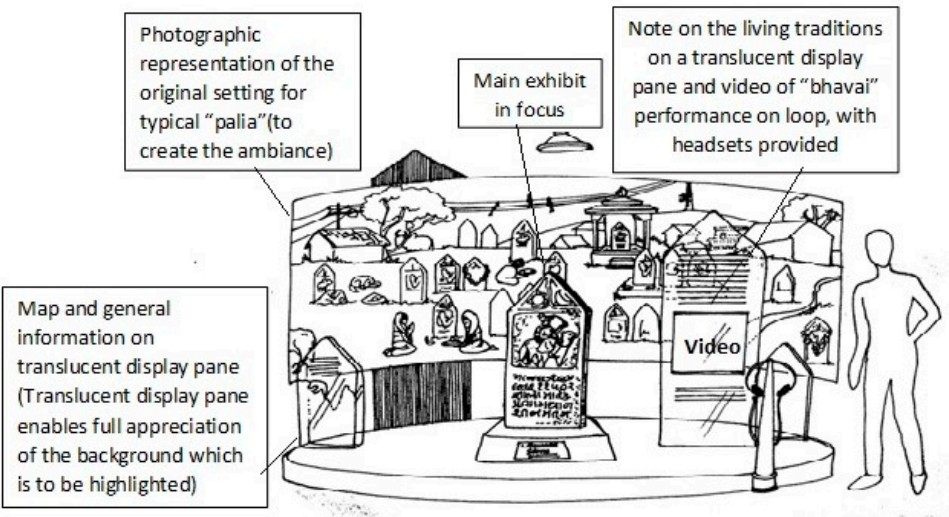

**Figure 6.** Illustration for a museum display of the memorials exhibiting audio-video data with the aim of simulating the cultural microcosm (conceptual sketch by the author).

The motivation for the current concern is the trove of oral narratives that lend the cultural literacy for the reception of the memorial stones. This provenance needs to be tapped into for an effective display and to convey the imposing presence of memorials in the sub-rural landscape in Kutch. A background image of memorials in a real setting from villages in Kutch will offer a peek into the sheer number of memorials that adorn the countryside. Once the memorials are situated in a simulated environment, the appreciation of the deified aspect can be enhanced with the audio-video inputs. Since memorials of this particular type—hero and *sati*—are a local tradition, unknown to parts outside Western India, Eastern Pakistan and some parts of Afghanistan, a universal appeal needs to be created for the display in a museum. The display will enable the storage and retrieval of oral narratives as provenance, and serve as datasets for researchers and Gujarati communities in diaspora.

## 6. Concluding Remarks

The study of the memorial stelae in Kutch through an active researcher-respondent interaction has opened new vista for probing into the provenance of otherwise lesser known origin stories and the seeds of a belief system. More importantly, the study suggests careful attention for the display of such cultural material that cannot be divorced from its locale. The best reception of artefacts such as the memorial stelae can be achieved through some modifications in display, such as recreating the surrounding performative tradition and simulating the setting in a landscape. The importance of creating such a sensorium has come to fore through this study on documenting oral narratives and the overall cultural literacy that enables appreciation of the cultural material.

This study has helped in accessing the potential for interdisciplinary data collection and analysis. The underused methodology of collecting oral narratives in India, and South Asia in general could aid in a better understanding of the cultural complex. An assessment and understanding of material cultural evidences beyond literary source analysis would be the way forward. Our data collection in the district of Kutch conducted through various means, which might be perceived as non-traditional, have yielded some previously unknown information about such artefacts which were always a commonplace in the local landscape of Gujarat. The living tradition surrounding the memorials could not have been otherwise tapped into if it was not for the systematic collection of oral reports from local respondents. The continued legacy of *bhavai* performances and palpable historical episodes reiterated through a material medium were studied through this documentation.

This case study will be indispensable for future documentation projects with conflicted provenance or multiple interpretations for a cultural object or artefact. The suggestions to disseminate the collected

data in the form of oral narratives will pave a way toward novel museum displays and retrieval systems combining tangible and non-tangible data-sets. Although not unique in the suggestion of display, the exhibition of culturally important objects such as these memorials, brings to the fore pertinent aspects of documentation for exhaustive comprehension. This case could be used as a prototype for other objects and artifacts of a similar nature, which warrants their presentation along with the cultural milieu in which they were originally set. This not only lends itself to better visitor reception but fosters detailed cultural data collection and justifies its display.

The oral legends and narratives appended to the datasets in discussion are invaluable sources of information about the artifacts, which are susceptible to loss and oblivion. Recording the oral legends about cultural objects such as these across a period of time will also enable the study of the trajectory of the re-description of regionally anchored notions. To extrapolate the case, one may use data such as these for social policy making in the future. Such culturally important data with cues for societal hierarchy, integration of culture, religion, and landscape will indeed add an important dimension to planning and research beyond the sphere of cultural studies.

**Funding:** A part of this research was funded by the Department of Archaeology, University of Kerala. No. Pl.A1/4288/Archaeology/13 dated 07-12-2015 Rs. 5000000 Documentation of Megalithic Sites in Kerala and Continuation of Excavations at Navinal in Gujarat and Explorations in Kachchh, Gujarat.

**Conflicts of Interest:** The author declares no conflict of interest. The field work and the credit for locating the memorials is to the entire team. This paper is based on the data from author's follow-up visits and further research.

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
