# Peer review of "Speaking Stones: Oral Tradition as Provenance for the Memorial Stelae in Gujarat"

_heritage, doi:10.3390/heritage2020071_

Round 1

Reviewer 1 Report

The paper presents a novel approach of investigating into a 'lesser explored' cultural phenomenon within the context of South Asia. Although a significant subaltern tradition, these memorials have rarely attracted focussed studies towards understanding the multifaceted nature of the tradition. Amongst its varied dimensions, studies on the oral narratives associated with the material relics are seldom undertaken, and as rightly pointed out by the author, have enormous potential towards understanding the significance of the memorials to their makers and the local populace who reside in their close vicinity in contemporary times. Further, it also highlights the role of collective memory seen through the associated intangibles. However, whilst accepting that the oral traditions to be invariably interpolated over a period of time, it is important to devise methodologies through which the subjective nature of such interpolations be eliminated. Such studies would invariably highlight the processes involved in glorifying and commemorating the deceased/heroic death, the sole purpose of erecting the memorial along with the shift in the 'identities' of the memorial over time. Whilst accepting the inherent difficulties in eliminating the subjective elements within oral traditions, a case of looking into patterns within oral traditions and other beliefs and practices is suggested. 

Besides the aforementioned observation, I observe a factual error in the paper, which needs correction, viz. the date for Pratimanatakan of Bhasa which reads as c.900 BCE (page 5 of the text). Also I recommend the author to use the broader term of 'South Asia' instead of 'India' to situate the geographical area, as these memorials are found distributed within the modern state of Pakistan as well.

Nevertheless, I appreciate the study undertaken by the author as it marks a new direction in the study of the memorial stone traditions in South Asia.

Author Response

Dear reviewer,

Thank you for your encouraging feedback.

I have rectified the date of Bhasa’s  Pratimanatakkam following your suggestion. The present draft of the  article has numbered sub-headings and some content is rearranged for  better flow of the content and development of the topic.

Thank you.

Warm Regards.

Durga

Reviewer 2 Report

This is an interesting topic being explored. Even though various data and ideas are stated, the paper is comprised from various sections largely unrelated to each other, as if being copied from a larger text, thesis or book. The lack of meaningful paragraphs and comprehensive sections renders the piece difficult to follow and detracts from possible merits. On the whole, it needs to be restructured to make specific points and propose solid arguments.

7: fieldwork as one word. Same across the text

10: to mesh together

12: Subject of the sentence is missing

14-5: Rephrase - elusive meaning

25: is full of instead of mushrooms

28: explain the rite of "sati"

38: Whose survey and fieldwork?

45-6: Syntax issues

52: But is this not the case with all or most of the ethnographic material exhibited in museums, i.e. that they are taken out of context just to be exhibited among irrelevant objects, sometimes from other cultural contexts?

52: inherent value? This needs to be further explained by relevant theoretical argument.

54: Again, are artefacts from precious material immediately more important/valuable than those from more humble materials?

62: Bands rather than registers

67: More details are needed on the nature of your study, the methodology followed and the results so far.

72: Explain the concept. Is this not the same as 'public good'?

76: There are no subheadings in the text, which makes it really difficult to read and understand. 

93: to be

99: Explain the social issues at Kutch. Why are the stelae important there?

100-110: Interesting point, but how this can be connected with the issues so far discussed? What is the role of these memorials inside and outside a community?

134: This is yet another use of the stelae, mentioned in passing here, not well examined or discussed.

145: Explain the template

148: Detailed study by whom?

153-173: This, although informative, looks as if it was copy-pasted from another piece, loosely connected with what comes before and after.

228: Merleau-Ponty

228: Various pieces of theory cited without proper explanation. 

237: Who is the team

253: This is part of the methodology and should go further up in the paper. 

281: A general discussion of the topics acknowledged in the oral tradition would have been interesting. 

282: Conclusions unrelated to discussion

Author Response

Dear reviewer,

Thank you for your comments. I benefitted greatly from your comments and suggestions for edits. The line numbers  in my responses are based on the current draft of the article submitted to MDPI.

7:

10: to mesh together

12: Subject of the sentence is missing

14-5: Rephrase - elusive meaning

25: is full of instead of mushrooms -is  dotted with

28: explain the rite of "sati" – lines 33-34

38: Whose survey and fieldwork? – lines 85-88

45-6: Syntax issues – have been worked on with the help from MDPI language service

52:  But is this not the case with all or most of the ethnographic material  exhibited in museums, i.e. that they are taken out of context just to be  exhibited among  irrelevant objects, sometimes from other cultural contexts? – Not to be taken as an attack on all ethnographic material at the museums,  but the case of the display of sati and hero memorials suffers  from this lack of attention. I think this is the result of acute neglect  for the study of these memorials.

52: inherent value? This needs to be further explained by relevant theoretical argument.- I was tempted  to add three studies of modern scholarship in this context, but I  suspected it to be too much of a distraction from the  main discussion. I might take this up in some other piece I write soon.  Thank you for your keen observation.

54: Again, are artefacts from precious material immediately more important/valuable than those from more humble materials?

62: Bands rather than registers - corrected

67: More details are needed on the nature of your study, the methodology followed and the results so far. – lines 70-131

72: Explain the concept. Is this not the same as 'public good'? – I take your point, and I  think it will be worthwhile to delve into some discussion in this  context. I have, for now just provided another  sentence in order to clarify.

76: There are no subheadings in the text, which makes it really difficult to read and understand. â€“ subheadings/section names are now in place

93: to be

99: Explain the social issues at Kutch. Why are the stelae important there? – I have not  touched on the social issues of Kutch, but have reiterated that the  memorials function as an indispensable part of the local  culture and a repository of local history.

100-110:  Interesting point, but how this can be connected with the issues so far  discussed? What is the role of these memorials inside and outside a  community? – some explanation in lines 180-195, 217-228

134: This is yet another use of the stelae, mentioned in passing here, not well examined or discussed. – added a line, probably basing the context better than before.

145: Explain the template – Some part in section 4 and conclusion

148: Detailed study by whom? – mentioned in methodology section

153-173:  This, although informative, looks as if it was copy-pasted from another  piece, loosely connected with what comes before and after.

228: Merleau-Ponty

228: Various pieces of theory cited without proper explanation. â€“ I acknowledge this  shortcoming, and I will take care from now on to maybe use limited ideas  in one piece without overburdening the  theoretical discussion which I cannot justify owing to word limit, etc.  

237: Who is the team

253: This is part of the methodology and should go further up in the paper. â€“ rectified in current version

281: A general discussion of the topics acknowledged in the oral tradition would have been interesting. 

282: Conclusions unrelated to discussion – I have added some content to this section.

I await your feedback, 

Warm Regards,

Durga 

Round 2

Reviewer 2 Report

Most of the points mentioned in my first review are covered. Good luck with the next steps of the research.

Author Response

Thank you for your comments. 

I have incorporated your suggestions and worked on section 5, to ground the content further.